# OpenReview forum: "In-Context Policy Iteration"
_ICLR.cc/2023/Conference — Submitted to ICLR 2023_

### Official Review · Reviewer_8XAi · 2022-10-23

**Confidence:** 2
**Clarity, Quality, Novelty And Reproducibility:** The paper writing is clear and the no…
**Correctness:** 4
**Technical Novelty And Significance:** 3
**Empirical Novelty And Significance:** 2
**Recommendation:** 6

**Strength And Weaknesses:**

Pros: (1) The paper is clearly written and easy to follow.

(2) The proposed method is novel in my opinion.

Questions: (1) In Alg. 1, can the authors specify what are prompt "parameters"? Are these the only trainable parameters?

(2) What will be the disadvantage of fine-tuning the foundation model parameters?

(3) Can the method work on more general control tasks such as mujoco?

**Summary Of The Paper:**

This paper introduces In-Context Policy Iteration that learns to perform RL with foundation models without expert demonstrations or gradients

**Summary Of The Review:**

The paper provides a simple idea of training RL with foundation models. But I'm not sure how previous related works conduct their experiments since the tasks seem very limited.

---

> ### Author Response · Authors · 2022-11-11
> **Response to reviewer questions**
>
> Thank you for your considered response. We have provided questions to your questions below.
>
> ## In Alg. 1, can the authors specify what are prompt "parameters"? Are these the only trainable parameters?
> The language in Alg 1 was confusing and has been changed. $\mathcal{D}$ is simply the replay buffer, the full history of (state, action, reward, termination) transitions that the agent has experienced.
>
> ## What will be the disadvantage of fine-tuning the foundation model parameters?
> Our algorithm targets a rapid learning regime in which tasks are mastered in hundreds of timesteps. Any gradient-based method is unlikely to have any benefit in such a short period of time.
>
> ## Can the method work on more general control tasks such as mujoco?
> We have added a section title “Limitations” addressing issues related to this question. In principle, the method can work on any control task with a discrete state space. All that the method requires is a sequence prediction model (in the paper we use Codex) capable of inferring state transitions and action probabilities given a history of behavior. Such a model will ensure that the rollouts (described in Computing Q-values) are unbiased Monte Carlo estimates of value for the current policy. Given such estimates, the algorithm will implement policy iteration, a method which is proven to converge.
>
> The domains in our paper were limited by the abilities of Codex to accurately predict transitions and actions in more complex domains. As sequence models mature (as they already have since our submission), we anticipate that more domains will become tractable for ICPI.

---

> > ### Author Response · Authors · 2022-11-15
> > **Follow up on response**
> >
> > Can the authors provide any additional information or experimental results to address the reviewer's concerns? Thank you for your response.

---

> > > ### Comment · Reviewer_8XAi · 2022-11-25
> > > **Response to authors**
> > >
> > > I thank the authors for their clarification. Although the authors state how their approach can in principle work in more complex tasks, the scalability in practice is still not clear. I decide not to change my score.

---

> > > > ### Author Response · Authors · 2022-11-25
> > > > **Thank you**
> > > >
> > > > The authors thank the reviewer for the feedback and the thoughtful consideration of our work.

---

### Official Review · Reviewer_jeCL · 2022-10-24

**Confidence:** 4
**Correctness:** 3
**Technical Novelty And Significance:** 4
**Empirical Novelty And Significance:** 2
**Recommendation:** 5

**Clarity, Quality, Novelty And Reproducibility:**

The paper was clear and well-written.

The quality of the experiments was good up to the issues raised above.

The algorithm is novel and interesting.

No code was provided so it is difficult to say how reproducible the results are.

**Strength And Weaknesses:**

### Strengths

1. The paper presents a clever way to leverage LLMs to perform RL entirely from in-context learning without any gradients. This is (to my knowledge) novel and seems like it could open up a new direction of research in using LLMs for RL.

2. The paper is clearly written and experiments seem relatively thorough on the toy problems that were selected.

### Weaknesses

1. As the authors admit in the paper, the tasks presented are very small-scale. It is unclear to me what the key bottlenecks are in scaling this approach to larger tasks. It is clear that it is limited to finite action spaces and to the ability to encode the state space into simple Python code, but it does not seem too challenging to embed some more complicated environments in this format. The paper would benefit substantially from either (a) scaling up to harder tasks or (b) including a more detailed discussion of the bottlenecks that prevented scaling up in this project.

2. The gains over the baselines were not always very clear and it was unclear how the baselines were chosen. On the first point, it would also be nice to see a more quantitative comparison to the baselines along with the regret curves. On the second, in particular, since all the domains are so small-scale it seems that it would be reasonable to include a more rigorous tabular baseline than Q-learning, for example a tabular model-based algorithm with more rigorous count-based exploration could potentially be a more fair comparison (especially since ICPI is using the LLM as a world model).

3. There are no results that attempt to analyze what the algorithm is doing beyond plotting regret curves. It would be nice to either see (a) some more qualitative/exploratory results that explain what sorts of errors each of the considered methods makes in each task or (b) some theory about what is required from the LLM for ICPI to actually work.

4. It is unclear whether how the "hints" fit into the standard RL framework and how they can be fairly compared against baselines that don't have access to them. This is not a major issue as the paper does provide the ablation without the hints, but it does seem like a potentially important caveat to the results that they required such hinting.

**Summary Of The Paper:**

This paper introduces in-context policy iteration (ICPI) an algorithm that can perform model-based policy iteration by leveraging a pre-trained large language model (LLM). ICPI uses prompts to create both a world model and a rollout policy out of the same LM. Specifically, the world model prompt uses examples from a buffer of transitions from the real environment and the rollout policy. ICPI then uses Monte Carlo rollouts of the rollout policy in the world model to estimate a Q function that is used to choose actions in the real environment. The paper presents a variety of experiments on toy problems with small, finite action spaces and small state spaces (finite or easily discretized) demonstrating improvement over some baselines.

**Summary Of The Review:**

Overall, I think the core idea presented in the paper is interesting and could be valuable to the community. However, I do have some concerns about why some things were done the way they were, so I am rating the paper as a weak reject for now. If the authors are able to resolve my questions, I would consider raising my score.

---

> ### Author Response · Authors · 2022-11-11
> **Response to Reviewer**
>
> ## The paper would benefit substantially from either (a) scaling up to harder tasks or (b) including a more detailed discussion of the bottlenecks that prevented scaling up in this project.
> The authors have added a paragraph titled “limitations” directly addressing this issue. In principle, ICPI can work on any control task with a discrete state space. All that the method requires is a sequence prediction model (in the paper we use Codex) capable of inferring state transitions and action probabilities given a history of behavior. Such a model will ensure that the rollouts (described in Computing Q-values) are unbiased Monte Carlo estimates of value for the current policy. Given such estimates, the algorithm will implement policy iteration, a method which is proven to converge. The bottleneck is therefore the capacity of the model to accurately predict these transitions and action probabilities.
>
> The domains in our paper were limited by the abilities of Codex to accurately predict transitions and actions in more complex domains. As sequence models mature (as they already have since our submission), we anticipate that more domains will become tractable for ICPI.
>
> ## It would also be nice to see a more quantitative comparison to the baselines along with the regret curves.
> The authors are happy to include additional comparisons. What other comparisons might be most clarifying?
>
> ## A tabular model-based algorithm with more rigorous count-based exploration could potentially be a more fair comparison
> We have added results using count-based exploration to the appendix. Note that all count-based methods assume a finite state-space, an assumption which our method does not make. We also note that the “Matching Model” is a tabular method which uses modeling in a similar manner to ICPI.
>
> Though we do outperform the baselines, this paper is not attempting to advance a new state-of-the art-algorithm. Rather the focus is on an idea: how to use the kinds of general knowledge that steadily improving foundation models contain to implement a new kind of in-context policy iteration. It will take access to better models and much more computation to scale this idea to sufficiently complex domains where one might draw an interesting comparison against competitive baselines. We primarily present the baselines as a reference to help the reader gauge the difficulty of the tasks, some of which have been modified by the authors from more familiar versions.
>
> ## It would be nice to either see (a) some more qualitative/exploratory results that explain what sorts of errors each of the considered methods makes in each task or (b) some theory about what is required from the LLM for ICPI to actually work.
> As stated in an earlier response, ICPI requires the LLM to correctly infer transition and action probabilities given a history of behavior encoded in the prompts, as described in the paper. We will include a section in the appendix describing the failure modes of the algorithm in greater detail. Errors result when the model fails to make accurate predictions about transitions and actions. One common error is to inaccurately predict reward for a given state/action. This can happen when too few transitions have been observed by the agent for the model to accurately infer the environment dynamics or the current policy. It can also happen when the logic of the environment is difficult to infer, relying on complex logic that exceeds the capacity of this generation of language models.

---

> > ### Comment · Reviewer_jeCL · 2022-11-25
> > **Leaving score the same**
> >
> > On the main issue of scaling up ICPI, I think the original issue still stands. Just asserting that scaling the LM will scale the algorithm is not sufficient. There needs to be an explicit argument or empirical evidence that scaling the LM indeed helps scaling ICPI. Moreover, I am not convinced that this is the true bottleneck to scaling ICPI. It seems that the real issue is the requirement of creating a hand-engineered pseudocode interface to the environment. In principle, this could be done on much more complicated environments than those tried in the paper, but then there may also be issues of scaling with size of the state/action space and horizon of the problem.
> >
> >
> > The count-based baseline added to the appendix does not seem to quite resolve the issue. First, there is no comparison to ICPI, so it is difficult to tell what is going on. Second, it seems that exploration does not actually help in any of the domains, which is somewhat confusing. Either the rewards really are so well-shaped in these environments that exploration is not needed at all, or there is some sort of bug in the implementation. One potential issue is the use of a 1/N bonus instead of the standard and theoretically justified 1/sqrt(N), moreover it could be better to use a more sound algorithm like model based UCRL2, but I am not entirely sure what is causing the issue.
> >
> > Thanks for adding some more qualitative analysis.
> >
> > Overall, I think the rebuttal did not change my opinion and I will leave my score the same.

---

> > > ### Author Response · Authors · 2022-12-03
> > > **Thank you for the feedback**
> > >
> > > Apologies for the delayed response. We did find a bug in the count-based exploration implementation and we are posting updated results here: https://pasteboard.co/OIKQAM263rKa.png
> > >
> > > These results include ICPI for comparison and they also use the corrected $1 / (1 + \sqrt{N})$ bonus. Thank you for the correction! On several of the graphs, the ICPI line might seem quite short. Note that we were unable to train ICPI fully to convergence in several domains due to time constraints (because of the rate-limits, ICPI can take days to train, even on simple domains).
> > >
> > > We still do not see a significant improvement from count-based exploration on several of the environments. Moreover, ICPI continues to outperform the tabular baseline, even with the count-based exploration. Our analysis is that these are not domains that pose a significant exploration challenge, insofar as reward is relatively dense and behaving greedily with respect to a value estimate is usually beneficial.
> > >
> > > ICPI's primary advantage over the tabular baseline is its ability to generalize to unseen states and actions using language semantics. For example, in the chain domain, the model can extrapolate beyond the observed observed states of 0 ... 8.
> > >
> > > If you have access to the OpenAI Beta, you can view this example where the LLM extrapolates to an unseen state (11): https://beta.openai.com/playground/p/oQEzAzD40hGRFsJMTxXYmyMe?model=code-davinci-002
> > >
> > > Here is a screen recording in case you do not have access to the Beta: https://youtube.com/shorts/LFo2F07ebDw
> > >
> > > Similarly, here is an example where it extrapolates to state (-1): https://beta.openai.com/playground/p/83gV56jXgXfzNGoSZ7VU662n?model=code-davinci-002 https://youtube.com/shorts/DVpgACYhsos
> > >
> > > You can view the full prompt here: https://gist.github.com/ethanabrooks/5d4ac39e559dd50ce985dd4582d283b1
> > >
> > > More general empirical evidence of this is the capability of ICPI to learn a near-optimal policy in the point-mass domain, where in general, states do not appear more than once, since the state-space is continuous.

---

> ### Author Response · Authors · 2022-11-14
> **Addition of Error Analysis**
>
> The authors have added a section titled "Error Analysis" to the appendix. In this section, we look at specific rollouts for the Chain, Distractor Chain, and Maze environments which produced suboptimal actions. The most common errors occur when the rollout policy chooses suboptimal actions, yielding the same value estimate for all environment actions (i.e. $Q\left(s, \mathcal{A}(1)\right) = \dots = Q\left(s, \mathcal{A}(n)\right) = 0$). We hope that this section will also make the internal workings of ICPI more transparent.

---

> ### Author Response · Authors · 2022-11-14
> **Link to Count-Based Exploration baseline**
>
> In addition to posting the count-based exploration results in the appendix, we've provided this link for your convenience: https://pasteboard.co/u32830dpchA0.png
> On these simple domains, it does not seem like exploration helped. In settings this small, exploration can distract from pursuing rewarding policies and slow learning. Please let us know if we can provide any further experiments or details.

---

### Official Review · Reviewer_Kfri · 2022-10-24

**Confidence:** 2
**Clarity, Quality, Novelty And Reproducibility:** The paper is clear, and of high quali…
**Correctness:** 4
**Technical Novelty And Significance:** 2
**Empirical Novelty And Significance:** 2
**Recommendation:** 5

**Strength And Weaknesses:**

The strength of this paper lies in using an LLM without any training. This is also a weakness, at its present form, the paper feels not that significant.


**Summary Of The Paper:**

The paper implements an implementation of the policy iteration algorithm using trained LLM.


**Summary Of The Review:**

I enjoyed the paper a lot. The presented utilized an LLM and, without any training, is able to solve a few RL tasks. However, I have serious doubts about its significance. What important research questions is being answered? Further,  very simple environments are treated and even then, some tricks are required (i.e. ejecting information about reward). It does not feel like the method would scale to anything bigger?

I admit not being an expert in the field of LLMs and I'd be happy to reconsider my evaluation.

---

> ### Author Response · Authors · 2022-11-11
> **Response to Reviewer**
>
> ## What important research questions is being answered?
> The primary aim of our work is to introduce an algorithm for in-context reinforcement learning that exploits large scale sequence models as world models and policies, and to provide initial evidence that the idea can work by exploring the effectiveness of variants of the algorithm in small illustrative domains, using existing pretrained language models. The key research question is thus whether this algorithm is in fact viable.  Contained within this investigation, we pursued several specific questions, including:
> - What are effective ways to construct prompts from recent histories? To this end, we describe several techniques for prompt design which empirically improve the capacity of Codex to behave as a world-model and a policy.
> - How can Q-values be estimated (and improved) using a general-purpose language model? To this end, we demonstrate a technique using rollouts that yields sufficiently accurate estimates in simple domains that learning occurs.
> - How does the size of the sequence model affect performance? We compare a variety of publicly available language models and demonstrate that only the largest are capable of performing the kinds of inference that our method requires.
>
> ## It does not feel like the method would scale to anything bigger?
> The authors have added a paragraph titled “Limitations” directly addressing this issue. In principle, ICPI can work on any control task with a discrete state space. All that the method requires is a sequence prediction model (in the paper we use Codex) capable of inferring state transitions and action probabilities given a history of behavior. Such a model will ensure that the rollouts (described in Computing Q-values) are unbiased Monte Carlo estimates of value for the current policy. Given such estimates, the algorithm will implement policy iteration, a method which is proven to converge. The sequence model is therefore the limiting factor for the scale of results we presented here, not the ICPI algorithm itself.
>
> The domains in our paper were limited by the abilities of Codex to accurately predict transitions and actions in more complex domains. As sequence models mature (as they already have since our submission), we anticipate that more domains will become tractable for ICPI.
>
> ## General Response
> LLMs and related large models are an important advance in AI that is only accelerating. They contain quite general and comprehensive knowledge about the world, and exhibit interesting capacities for generalization. There is hope that such knowledge can be unlocked and used for RL tasks. Others have explored a few different ideas, for example, Huang et. al. demonstrated the use of LLMs for planning, given expert-designed prompts. Others, like Chen et. al. trained new models on RL data from the downstream task and demonstrated the capacity of these models to produce good policies and even outperform the source algorithm in some instances. We show that a surprisingly simple idea can allow general-purpose foundation models to solve RL tasks without any assumptions about the downstream task. Our experiments were indeed simple because we only had limited access to an early LLM (we relied on the free Codex beta which permits only 20 queries per minute). We expect that as these models improve, ICPI will become more practical and useful.

---

> > ### Comment · Reviewer_Kfri · 2022-11-24
> > **Thank you**
> >
> > I thank the authors for the answers. I can see some valid points. I acknowledge the scientific value points. I am still not convinced about scalability, though mapping from the states to their meaningful textual representation seems obscure to me.
> >
> > I raise my score.

---

> > > ### Author Response · Authors · 2022-11-24
> > > **Thank you**
> > >
> > > Thank you for taking our points into consideration. Regarding the textual representation, this is indeed a scalability bottleneck. However, the ICPI algorithm is also applicable to other kinds of sequence models. Consider recent work on video models, which would require some visual representation of the environment, a format that seems much more amenable to scale. Also consider work on "RL models" like Gato, which process raw RL data, and would require no re-representation. As work on sequence modeling continues to  develop, we conjecture that this bottleneck will no longer be an issue.

---

> ### Author Response · Authors · 2022-11-14
> **Follow-on response**
>
> Please let us know if we can provide further information or experimental results that time will allow. We look forward to your reply.

---

> ### Author Response · Authors · 2022-11-17
> **Follow up**
>
> There is only one day remaining in the discussion period. Please let us know if we can provide any further information or experimental results. Thanks.

---

### Official Review · Reviewer_wJ8R · 2022-10-24

**Confidence:** 4
**Correctness:** 3
**Technical Novelty And Significance:** 3
**Empirical Novelty And Significance:** 3
**Recommendation:** 6

**Clarity, Quality, Novelty And Reproducibility:**

**Clarity:** For the most part, the paper is written clearly and the motivation is laid out very well. Perhaps adding further details to the descriptions outlining how $D_b$, $D_r$, $D_s$ and $D_a$ are picked during rollouts would help.

**Novelty:** While the paper doesn't represent a conceptual leap (i.e. the ingredients of ICLI were present at the type of publication), the execution and precise formulation appear to be novel.

**Reproducibility:** While the proposed algorithm is clear overall, producing a faithful replication of the work seems difficult (i.e. how the $D_i$ prompts are constructed is not extremely clear, etc.). There's also no released codebase.


**Typos:**
* Citep should be used when citing Codex in the abstract.
* While described, $D_s$ is not mentioned in the second paragraph in the "Computing Q-values" section.
* In the "prompt-format" section, the sentence "Note that while hints are provided hints in the initial context, [...]"
* In "comparison of ICPI with baseline algorithms:" "This baseline assumes access **to** a [...]".




**Strength And Weaknesses:**

**STRENGTHS**:
* **No need for expert demonstrations:** The proposed algorithm doesn't require expert demonstrations, or any form of imitation learning.
* **No weight updates:** There's no need to compute gradients through the LLMs during learning: only (discrete) prompts are refined over training to facilitate learning.
* **Task difficulty appear well-chosen:** While relatively easy from an RL perspective, the six tasks presented in the paper appear to capture the right difficulty to test ICPI. Only Codex seems to get all tasks right, which gives useful signal as to what about the LLMs could be responsible for the reported performance. The chosen tasks are also in increasing complexity, and each display qualitatively different abilities.
* **Nice variability of LLMs:** The authors seem to cover a sufficient range of publicly available LLMS.

**WEAKNESSES:**
* **Necessity of hints and tweaks:** It appears that for any sufficiently nontrivial environment, human-engineered hints are needed to get ICPI to work. The fact that some statistical rebalancing (of $D_r$ and $D_r$) also signal brittleness (which might potentially disappear as the LLMs get better).
* **Restricted to text domain:** Currently, ICPI is limited to domains where LLMs can exploit their pretrained knowledge (whether it be natural language or code).
* **Issues regarding scaling:** The proposed method might have difficulty scaling to tasks with larger difficulty than those explored in the paper, as prompting LLMs with a large number of "state-action-termination" tuples when the dimensionality of these are high might become computationally challenging.

**QUESTIONS TO THE AUTHORS:**
* **Do LLMs play a unique role in this algoritm?** I wanted to ask further questions about the precise role LLMs play in the proposed algorithm. There doesn't appear to be an explicit pressure on the LLM to seek actions that yield higher returns (which could have been there if weight updates were allowed, or if it were prompted with expert trajectories, or perhaps the LLM were prompted with an explicit instruction to seek higher rewards). Instead, the algorithm seems to improve over time as the LLM is prompted with data that's increasingly coming from better (i.e. higher reward) rollouts. If this is the case, why would an LLM be necessary? Even if one randomly sampled termination, reward, next state and actions from $D_b$, $D_r$, $D_s$ and $D_a$ during rollouts, wouldn't learning still occur? What exactly is the benefit of using an LLM here? Perhaps the pretraining data helps pick better actions in a nontrivial way?
* **Is Tabular Q tuned well?** All of the tasks seem simple enough that the Tabular Q algorithm should eventually learn the optimal policy (is this right?) However, that doesn't appear to be the case for certain environments. What seems to have gone wrong? Perhaps tabular-Q takes a bit longer to converge? Or perhaps the hyperparameters (most importantly the learning rate) aren't tuned separately for each task?
* **Request for additional baselines and ablations:** At each rollout step, the LLM gets queried 4 times. Time permitting, could you please run an ablation where you systematically replace each one with a random sample from $D_x$ where $x \in \{b, r, s, a\}$ while keeping the other three samples intact (i.e. they still use LLM)? I'm asking this to see if all of these queries are similarly important for the apparent superiority of ICPI over other baselines such as the "matching model baseline".
* **Model outputs during the first few rollouts:** Since $D$ is empty at the beginning of training, it's not clear how the LLM should be prompted to get it to output sensible things. Is this every a problem? How do you address it?
* **How exactly are $D_i sampled?** The text mentions that "$D_b$ contains [...] tuples sampled randomly from the $D$", Do you mind explaining how this exactly works? Is random sampling done uniformly? How many samples are picked?
* **Further details on the Point-mass task:** Do you mind explaining what "accel(pos, vel)"does exactly?


**Summary Of The Paper:**

The authors propose In-Context Policy Iteration, a Q-learning like algorithm that iteratively prompts large language models (LLM) with trajectories that have yielded better rewards and taps into the in-context learning abilities of LLMS to simulate policy rollouts. The proposed algorithm can self-improve without weight updates or expert demonstrations.

At its core, ICPI queries an LLM using past trajectory data (i.e. replay buffer) to obtain simulated rollouts that are used to estimate Q values, which are then used to take actions and augment the replay buffer.

Contributions:
* The authors propose the ICPI algorithm, which enjoy the properties described above.
* They evaluate it's performance on 6 toy tasks and report results on:
  * a few sensible baselines,
  * algorithm ablations, which involve some tips/tricks to get ICPI working, and
  * the performance on ICPI using different core LLMs, including OpenAI's Codex.

**Summary Of The Review:**

The paper proposes ICPI, an algorithm that enables LLMS to learn to solve simple RL tasks without any expert demonstrations and weight updates.

While there are concerns related to the broad applicability and scalability of this algorithm, I believe that the novelty and scientific value of the results justify acceptance.

---

> ### Author Response · Authors · 2022-11-11
> **Response to reviewer questions**
>
> ## Do LLMs play a unique role in this algorithm?
> An LLM (or other sequence model) is necessary to predict states/actions/reward given a history of behavior. In the case of predicting observations, for example, this requires the model to infer environment dynamics from the prompt and apply these dynamics to the current observation. The better the model generalizes to new states, the quicker the algorithm will learn.
>
> ## Is Tabular Q tuned well?
> We have posted learning curves for our hyperparameter sweep in the appendix. The algorithm converges for all environments except point-mass but takes longer than ICPI. Note that the state space of point-mass is continuous and therefore intractable for Tabular Q.
>
> ## Request for additional baselines and ablations:
> We are currently running these experiments on the chain domain and will share the results as soon as they are finished. Because of the rate-limit for the Codex API, runs can take several days.
>
> ## Even if one randomly sampled termination, reward, next state and actions from $\mathcal{D}_b$, $\mathcal{D}_r$, $\mathcal{D}_s$ and $\mathcal{D}_a$ during rollouts, wouldn't learning still occur?
> The problem with randomly sampling termination, reward, states, and actions is that the predictions, and therefore the value estimates, are unlikely to be accurate. This will make it difficult for ICPI to improve the agent’s behavior.
>
> ## What exactly is the benefit of using an LLM here? Perhaps the pretraining data helps pick better actions in a nontrivial way?
> We did not observe any evidence that the pretraining data inclined the LLM to pick better actions, but also note that  the method does not require this. Codex seems to treat the words in the prompt primarily as symbolic values and completes patterns with no discernible reference to their semantic value. The exception to this is that Codex does understand concepts like: 4 is less than 5 and left is opposite from right. This helps Codex generalize in some cases to novel states and actions.
>
> ## Model outputs during the first few rollouts
> When D is empty, the LLM receives only the “query,” e.g. the current observation in the case of action prediction. In general this yields inaccurate predictions and random behavior. The method nevertheless works because once experience accumulates, even from random behavior, the model can start making predictions about states and rewards  and these can be used to estimate values for the current (random) policy using rollouts. Once we have value estimates, we can improve the policy using the $\arg\max$.
>
> ## How exactly are $\mathcal{D}_i$ sampled?
> We have revised the description of this in the paper in order to improve clarity. We provide a description here as well.
> Recall that the replay buffer contains the full history of (state, action, reward, termination) tuples, each corresponding to a single timestep of agent behavior. In order to construct $\mathcal{D}_i$, we scan through the buffer and filter out tuples that do not meet a set of constraints which vary per prediction type. Once we have collected the tuples that meet these constraints, we shuffle them, convert them to strings, and clip whatever exceeds the 4000-token context limit. Constraints are as follows:
> - For state predictions: we filter out terminal states, since the model never needs to make a next-state prediction for a terminal state, and constrain the action to match the action for which the model is making a prediction. For example, if we are predicting the next state given a “left” action, we keep only timesteps for which the action is “left.”
> - For reward predictions: we constrain the action to match the current action. If the model predicts that the current state is terminal, we filter out non-terminal timesteps. Similarly if the current state is nonterminal, we filter out terminal timesteps. Finally we balance the number of timesteps corresponding to each reward value. For example, if there are 3 timesteps with reward of 1 and 13 with reward of 0, we repeatedly choose a random timestep with reward of 1 and duplicate it until there are 13 timesteps with reward of 1.
> - For termination prediction, we also constrain the action to match the current action. We balance the number of terminal and nonterminal timesteps using the duplication method described in the previous bullet.
>
> ## Further details on the Point-mass task:
> We have added additional clarification to the main text. Here is an example that illustrates the domain:
> `accel(pos, vel)` moves the position by the amount `vel` and increments `vel`. For example if `pos` were 3 and `vel` were -2, `accel(pos, vel)` would decrease `pos` to 1 and increase `vel` to -1
>
> ## Reproducibility:
> Here is a link to an anonymized github repository: https://anonymous.4open.science/r/icpi-5812.
> A non-anonymized link will be posted in the camera-ready version of the paper.
>
> ## Typos:
> Thank you for noting these. These are fixed in the revised version.

---

> > ### Comment · Reviewer_wJ8R · 2022-11-15
> > **Thank you for your response**
> >
> > Thank you for your response and clarifications.
> >
> > I want to confirm something about the ablation results shared via the png: 1) When sampling randomly, you did sample from D_b ... D_a, right (as opposed to literally randomly sampling)?
> >
> > Also, it's still not quite clear what kind of knowledge that's present in the context that LLMs are making use of. Provided that the environments that LLMs are evaluated on weren't in the pretraining data (which the authors also argue), the most plausible explanation seems something akin to the following: "LLMs are performing some kind of dictionary lookup-like operation that lets them pick actions. Since over time, the context is more likely to be filled with samples from trajectories that lead to high rewards, this lookup often also picks decent actions". Do authors agree with this characterization? What else about using LLMs enables the algorithm to work? Adding a discussion on this to the paper would improve the paper, in my opinion.

---

> > > ### Author Response · Authors · 2022-11-15
> > > **Reply**
> > >
> > > ## Response to 1)
> > > It is possible that the authors misunderstood the baseline that you had in mind. We did indeed sample randomly from values within the domain of the prediction category. For example, we sampled actions from the set of valid actions and then encoded them as strings. Perhaps what you had in mind was to take, for example, one action that was present in the prompt (e.g. present in $\mathcal{D}_a$ and use that in place of the action predicted by the LLM. If this is correct, We can run this baseline and report back as soon as it is complete. Note that this is quite similar to the "Matching Model" baseline which uses the most recent transition in the prompt.
> > >
> > > ## Whether LLMs are performing dictionary lookup
> > > There are cases when the LLM does indeed retrieve exact values present in the prompt. However, the LLM is also able to infer patterns and thereby generalize beyond the prompt. For example, the LLM can extrapolate beyond the observed observed states of `0 ... 8`.
> > >
> > > If you have access to the OpenAI Beta, you can view this example where the LLM extrapolates to an unseen state (11): https://beta.openai.com/playground/p/oQEzAzD40hGRFsJMTxXYmyMe?model=code-davinci-002
> > >
> > > Here is a screen recording in case you do not have access to the Beta:
> > > https://youtube.com/shorts/LFo2F07ebDw
> > >
> > > Similarly, here is an example where it extrapolates to state (-1):
> > > https://beta.openai.com/playground/p/83gV56jXgXfzNGoSZ7VU662n?model=code-davinci-002
> > > https://youtube.com/shorts/DVpgACYhsos
> > >
> > > You can view the full prompt here:
> > > https://gist.github.com/ethanabrooks/5d4ac39e559dd50ce985dd4582d283b1
> > >
> > > More general empirical evidence of this is the capability of ICPI to learn a near-optimal policy in the point-mass domain, where in general, states do not appear more than once, since the state-space is continuous.

---

> > > > ### Author Response · Authors · 2022-11-15
> > > > **Update on Baseline**
> > > >
> > > > The corrected baseline is running now. We will report back with results as soon as we have them.

---

> ### Author Response · Authors · 2022-11-14
> **Addition of "Random Prediction Ablation."**
>
> A new section titled "Random Prediction Ablation" has been added to the Appendix. Unfortunately, due to time constraints, the authors were only able to run the ablation on the Chain environment. However, none of the ablations were able to learn in this setting. It is unlikely that their performance would improve on the more difficult environments.
>
> The figure may also be viewed at this link: https://pasteboard.co/yKXrrgPwHZNV.png

---

> ### Author Response · Authors · 2022-11-14
> **Link to Tabular Q hyperparameter search**
>
> As indicated in our earlier message, we have included hyperparameter sweeps of the Tabular Q baseline in the appendix. For your convenience, we have also posted them at this link: https://pasteboard.co/u32830dpchA0.png

---

> ### Author Response · Authors · 2022-11-17
> **Follow up on baseline**
>
> The baseline you requested is still running, but we are posting the partial results in order to allow time for a response:
> - The graph can be viewed here: https://pasteboard.co/8ih068GWA9mw.png
> - For $\mathcal{D}_s$ and $\mathcal{D}_r$, we filter for past timesteps with the same (state, action) as the current timestep. I thought this was implicit in your suggestion. Without this, the predictions retrieved from the buffer would be no good.
> -  For $\mathcal{D}_a$, we filter for past timesteps with the same state as the current timestep.
> - When no past timesteps are found that meet the aforementioned criteria, we terminate the rollout and use the partial trajectory for our value estimation.
> - Your baseline would perform quite well on a small domain like Chain, where the agent is likely to experience all possible transitions early in training. Since time was short and we had to choose a single domain, we chose Mini-Invaders, a domain with higher state complexity, which would demonstrate some advantage for our method.
> - Examining the performance of your baseline on this domain, we observe that performance drops when the agent encounters unseen transitions, whereas ICPI is usually capable of handling these cases, due to the generalization capabilities of the LLM. This accounts for ICPI's current advantage over the baseline.

---

> ### Author Response · Authors · 2022-11-18
> **Baseline final result**
>
> The final result for the requested baseline can be viewed here: https://pasteboard.co/ZYAwnoVQfwvK.png
>
> The only follow-on to our previous analysis is that the $\mathcal{D}_a$ baseline was the most competitive, which makes sense, since in many domains, policy iteration can be effective even with value estimates based on a random policy.

---

> > ### Comment · Reviewer_wJ8R · 2022-11-26
> > **Thank you for your response.**
> >
> > Thank you for your response.
> >
> > This graph makes more sense than the previous one! It looks like randomized selection indeed turned out to be a strong baseline, though on the task evaluated, it looks like the LLM approach still is better. I believe tuning this baseline for the other tasks for camera ready would significantly strengthen the scientific contribution of the paper.
> >
> > I'll maintain my score for now (leaning more towards accept) pending reviewer discussion.

---

> > > ### Author Response · Authors · 2022-12-03
> > > **Thank you for the feedback**
> > >
> > > We will certainly include this baseline in the camera-ready version of the paper.

---

### Decision · Program_Chairs · 2023-01-20

**Decision:**

Reject

**Justification For Why Not Higher Score:**

See the aforementioned major concern and breach of anonymity.

**Justification For Why Not Lower Score:**

N/A.

**Metareview: Summary, Strengths And Weaknesses:**

The main contribution of this work lies in the proposed in-context policy iteration algorithm that exploits pretrained large-language models for solving small, simple RL tasks without needing expert demonstrations nor weight updates, which is interesting, new, and can potentially pave the way for future works.

After reviewing and responding to the authors' rebuttal, a major concern that is raised by all reviewers is that the empirical evaluation of the proposed approach is based on small simple RL tasks and it is unclear whether it can scale well to more complex tasks. We acknowledge that the authors have responded to this concern by saying that their approach can scale in principle, but the main culprit for this limitation is the large language model being used at this time. They believe that this concern will be resolved with the advancement of LLMs in the future. Their current work is simply to show its feasibility, albeit in small simple RL tasks. Such a response has drawn mixed feedback from the reviewers: On one hand, the authors have indeed demonstrated its feasibility through simple RL tasks. On the other hand, it is unclear whether their approach can indeed scale in practice.

We strongly encourage the authors to revise their work based on the reviewers' feedback.

Unfortunately, the authors have breached anonymity in their revised paper. Hence, we regrettably cannot recommend accepting this work.

**Summary Of Ac-Reviewer Meeting:**

There is sufficient **written** discussion generated on the OpenReview discussion forum to the extent of being able to reach a consensus on the recommendation. Hence, there is no need for a meeting.

Furthermore, the authors have breached anonymity in their revised paper.